# Surface-Engineered Extracellular Vesicles in Cancer Immunotherapy

**DOI:** 10.3390/cancers15102838

**Published:** 2023-05-19

**Authors:** Vinith Johnson, Sunil Vasu, Uday S. Kumar, Manoj Kumar

**Affiliations:** 1Department of Chemical Engineering, Indian Institute of Technology, Tirupati 517619, India; 2Department of Radiology, Stanford University, Stanford, CA 94305, USA

**Keywords:** extracellular vesicles, exosomes, microvesicles, apoptotic bodies, surface engineering, tumor-secreted EVs, tumor immune microenvironment, drug delivery system, immunomodulation, cancer immunotherapy

## Abstract

**Simple Summary:**

Extracellular vesicles are small membranous particles secreted by cells. Extracellular vesicles facilitate the transportation of biomolecules, such as protein, RNA, and DNA fragments, to communicate with neighboring and distant cells. Cancer cells use extracellular vesicles to hijack the immune system and induce cancer-promoting signals. Modifying extracellular vesicles using surface engineering tools allows the addition of biomolecules for targeted delivery, thus modulating the hijacked tumor immune microenvironment to improve therapeutic efficacy. This review article discusses extracellular vesicle modification strategies explicitly focusing on the approaches used for surface engineering. We revisit the work carried out on the surface-engineered extracellular vesicle and its application in immunomodulating tumor microenvironments for cancer immunotherapy.

**Abstract:**

Extracellular vesicles (EVs) are lipid bilayer-enclosed bodies secreted by all cell types. EVs carry bioactive materials, such as proteins, lipids, metabolites, and nucleic acids, to communicate and elicit functional alterations and phenotypic changes in the counterpart stromal cells. In cancer, cells secrete EVs to shape a tumor-promoting niche. Tumor-secreted EVs mediate communications with immune cells that determine the fate of anti-tumor therapeutic effectiveness. Surface engineering of EVs has emerged as a promising tool for the modulation of tumor microenvironments for cancer immunotherapy. Modification of EVs’ surface with various molecules, such as antibodies, peptides, and proteins, can enhance their targeting specificity, immunogenicity, biodistribution, and pharmacokinetics. The diverse approaches sought for engineering EV surfaces can be categorized as physical, chemical, and genetic engineering strategies. The choice of method depends on the specific application and desired outcome. Each has its advantages and disadvantages. This review lends a bird’s-eye view of the recent progress in these approaches with respect to their rational implications in the immunomodulation of tumor microenvironments (TME) from pro-tumorigenic to anti-tumorigenic ones. The strategies for modulating TME using targeted EVs, their advantages, current limitations, and future directions are discussed.

## 1. Introduction

Extracellular vesicles (EVs) are small membrane-bound particles that mediate cell-to-cell communication. All cells secrete EVs generated via intrinsic cellular biogenic pathways as the cargo of signaling molecules typically involved in cell regeneration, differentiation, and proliferation [1]. As small lipid bilayer particles, EVs carry surface proteins, lipids, cytokines, glycans, and other encapsulated molecules [2]. Apart from bioactive molecules, EVs can also encapsulate various cellular organelles, including the transfer of functional mitochondria to other cells, and promote cell survival and tissue regeneration [3]. The composition of each EV subpopulation differs based on their formation mechanisms. These EVs are broadly categorized into exosomes, microvesicles, and apoptotic bodies, ranging between 30–5000 nanometers (nm) [4]. Exosome biosynthesis involves an endosomal sorting complex required for transport (ESCRT), which guides intraluminal vesicle transport within late endosomes to the plasma membrane. The matured endosomal membrane fuses with the plasma membrane and causes the exocytosis of exosomes in the size range of 30–150 nm in diameter [5]. Unlike exosome formation, microvesicle formation does not require exocytosis. Microvesicles are solely derived from the budding of plasma membranes under external stimuli or stress factors. The membrane composition of microvesicles closely represents that of the plasma membrane of parent cells [6]. On the contrary, apoptotic bodies manifest from membrane blebbing, which involves the formation of small protrusions or blebs of the plasma membrane [7]. The membrane and luminal compositions of apoptotic bodies have distinct characteristics. Apoptotic bodies are enriched with phosphatidylserine (PS) phospholipid on the outer leaflet as opposed to their presence in the inner leaflet in the plasma membrane of a healthy cell [8]. PS on the outer leaflet of apoptotic bodies serves as an “eat me” signal and facilitates recognition by the macrophages for disposal. 

The cellular uptake mechanism of EVs depends on the recipient cells’ membrane receptor components. The surface proteins and receptors facilitate the initial adhesion of exosomes to recipient membranes [9]. Upon adhesion, the membrane-associated exosomes are usually endocytosed via phagocytosis, micropinocytosis, or clathrin/caveolae-mediated endocytosis, trigger the transduction of intracellular signaling pathways or integrate with cell membrane-transferring protein for cell internalization. Microvesicles can also internalize via surface receptors or ligands, including integrins, ligands such as tetraspanins (CD63, CD9, or CD81), and heat-shock proteins. These receptors are core protein signatures of EVs and play a prominent role in cellular uptake [10]. Since microvesicles exhibit a broad range of size distribution (50 nm to 5 µm), their uptake mechanism is also determined, to a certain extent, by their sheer size. The membrane of apoptotic bodies carries a significant extent of tumor-associated macrophage (TAM) receptors, phospholipids, integrins, and scavenger receptors that enable the specific recognition of apoptotic bodies by immune cells such as macrophages and dendritic cells. Their internalization by cells usually culminates in lysosomal degradation [11]. 

Tumors deploy EVs to alter stromal cell behavior and promote metastasis. EVs are increasingly shown to be involved in the invasive–metastatic cascade. Tumors secrete a heterogenous population of EVs to shape the tumor microenvironment (TME) and change its malignant behavior in response to immune surveillance or therapies [12]. EVs could play the role of a critical biomarker in the diagnosis and prognosis of cancer [13]. While the current understanding of spatial interactions between the host and tumor cells remains poor, EV-mediated signaling has provided critical insights into these interactions. Malignant cells secrete EVs to promote angiogenesis and modulate the immune system to support cancer progression [14,15,16,17]. EVs isolated from cancer patients have been associated with metastasis or relapse [13]. Tumors orchestrate stromal cells via EV-mediated systemic reprogramming and support pre-metastatic niche formation and subsequent metastasis [12,18]. This allows cancers to differentially exhibit the temporal course of proliferation and the metastatic progression of distant organs.

Surface-engineered EVs allow preferential internalization through one mechanism over another when modified to target a specific signaling pathway. The membrane proteins and lipids modifications can achieve cell-specific delivery or interactions with great promises in cancer immunotherapy [19]. The International Society for Extracellular Vesicles (ISEV) established guidelines for the separation and characterization of extracellular vesicles (EVs) [20]. These guidelines provide protocols for EV isolation, characterization, and functional analysis to ensure reproducible and comparable results from EV research labs. However, some studies conducted before establishing the ISEV guidelines may not have adhered to these standards. As a result, it is challenging to distinguish the nature of extracellular vesicles reported in the pre-existing literature from before 2014 [21]. Considering this ambiguity, we have structured this review article based on surface modification strategies rather than on the kind of extracellular vesicles. Just like variations in the membrane components of exosomes, microvesicles, and apoptotic bodies, their luminal cargo composition also differs to a great extent. A detailed review of the luminal cargo composition of the EV has been discussed in great depth elsewhere [22]. Here, we focus on EV membrane engineering and compositions and their application in cancer immunotherapy. We revisit recent advances in surface-engineered EVs applying genetic engineering, chemical conjugation, and lipid insertion strategies for their application in cancer immunotherapy. 

## 2. Surface Engineering of EVs

EVs have a unique origin and molecular composition that renders them highly stable in the bloodstream; thus, they have potential in cancer immunotherapy [23]. However, their efficacy is hampered by ineffective tumor targeting and many surface modification strategies have been implemented to improve tumor targeting (Figure 1) [24]. These strategies enhanced targeting specificity and therapy by introducing new functionalities via a targeted ligand and therapeutic molecules [23]. Precise surface engineering of EVs can be achieved via genetic engineering tools controlling cellular biosynthesis pathways. The two primary tumor immunomodulatory approaches used in this strategy are the modification of EV surfaces to express tumor-specific antigens or immune checkpoint inhibitors to recruit the immune cells [25]. Unlike the genetic engineering approaches, chemical conjugation involves attaching specific molecules or ligands to the surface of EVs using chemical reactions. These molecules could be antibodies, peptides, or other targeting ligands that selectively bind to cancer cells or immune cell receptors to enhance therapeutic efficacy [26]. Lipid insertion is another promising approach for engineering EV surfaces for cancer immunotherapy [27]. Unlike the targeted antibodies and peptides involved, lipid-insertion molecules are generally immunostimulatory, play a role in the stability and longevity of vesicles, and, at times, act as therapeutic agents [28]. The unique advantage of lipid insertion over chemical conjugation is that it allows the precise control of the density and orientation of the inserted molecules on the EV surfaces, which can optimize their targeting and therapeutic efficacy. The preferential interaction of lipids with immune cells acts as a double-edged sword; it comes at the cost of possible toxicity and acute immunogenicity if the lipid insertions are not well-optimized [29].

## 3. Genetic Engineering of EV Surfaces for Cancer Immunotherapy

As a transport cargo, the intrinsic property that allows EVs to bypass immune surveillance favors EV fabrication. Cell genetic engineering allows the obtention of desired EV functions to target specific cell types [30]. A good example is using EVs for the controlled activation of immune cells, such as dendritic cells, T-cells, and natural killer cells, in the immunomodulation of TME. Engineered EVs shed from parent cells could improve the specific target recognition efficiency, targeting ability, and anti-tumor efficacy [31,32,33]. Thus, EV surface modification provides a promising clinical application [34]. Typically, the genetic modification of the EV surface is achieved via transfection and activation through protein expression, via tagging molecules on the surface protein and DNA or via RNA delivery to TME [35]. The surface-engineered exosomes with aptamer-based DNA nano assemblies have benefited theragnostic applications [36]. For example, EVs were engineered to express Toll-like receptor (TLR) agonists or co-stimulatory molecules as immunostimulatory proteins [37]. These modifications enhance EVs’ ability to induce a robust anti-tumor response via immune activation in another study [38]. The study demonstrated EV-based activation of dendritic cells and a robust anti-tumor immune response being induced in mice. Thus, using EVs as a delivery system molecule holds great promise for effective cancer immunotherapies.

### 3.1. EVs Carrying Immune Receptor and Ligand Protein

Another EV surface modification strategy is attaching monoclonal antibody-derived chimeric antigen receptors (CARs) for specific binding on cancer cells [39]. CARs are synthetic proteins that can specifically recognize and bind to tumor cell-surface antigens [40]. Engineered EVs bind to specific receptors on tumor cells when engineered to express CARs, deliver immunomodulatory proteins to target cells, and induce an immune response against cancer cells. For example, Shi et al., 2020, engineered EVs to express a human epidermal growth factor receptor-2 (HER2)-specific CAR against HER2 protein overexpressed in breast cancer [41]. This study demonstrated CAR-EVs’ potential as novel cancer immunotherapy using in vitro and in vivo breast cancer models. Likewise, Fu et al., 2019, demonstrated that exosomes derived from CAR-T-cells have potent anti-tumor effects and low toxicity [42]. The exosomes produced by the engineered CAR-T-cells also expressed the same chimeric antigen receptor as the parent CAR-T cells did. The study found that these CAR-exosomes could target and kill cancer cells with a much lower toxicity profile than CAR-T-cell therapy [42]. Therefore, CAR-exosomes could be a promising alternative to CAR-T-cells in cancer immunotherapy. This approach has been reviewed and extensively discussed in a recent review article by Pagotto et al., 2023 [40]. EVs naturally contain membrane-associated immunoregulatory molecules, including the immune-checkpoint molecules such as programmed death ligand s1 (PD-L1), cytotoxic T lymphocyte antigen-4 (CTLA-4), and the apoptosis-inducing ligands FASL and TNF-related apoptosis-inducing ligand (TRAIL). These immune checkpoints enable EVs to interact with cognate ligands and receptors expressed by T-cells and natural killer (NK) cells in TMEs [43]. An alternative strategy with which to modify the surface of EVs for targeting specific immune cells or tumor cells is the incorporation of peptides, antibodies, or ligands [44]. This approach can also deliver therapeutic cargo, such as RNA and chemotherapy drugs, to specific cells.

The surface engineering of exosomes by the aptamer-based DNA nano assemblies has significantly benefited the theragnostic applications (Figure 1) [36]. TRAIL is a therapeutic agent which induces apoptosis via targeting the death receptors 4 and 5 on cancer cells [45,46]. The TRAIL can be loaded onto the cells via transfection [47,48]. TRAIL-containing exosomes were developed by transducing K562 leukemic cells with the TRAIL lentivirus expression vector [47]. The secreted exosomes exhibited the enhanced apoptosis of lymphoma and melanoma cells. Additionally, exosomes with TRAIL were created using engineered mesenchymal stem cells (MSCs), resulting in the apoptosis of various cancer cell lines, including breast, renal, lung, and mesothelioma [48]. 

One of the most widely used exosome surface proteins to display a targeting motif is LAMP-2B, a lysosome-associated membrane protein (LAMP) member. LAMP-2B is predominantly available on the lysosomes and endosomes, and a smaller fraction is expressed on the cell surface. The abundant expression of the LAMP-2B protein was reported on the dendritic cell-derived exosomes [19]. LAMP-2B, present on the surface of the exosomes, is a conventional site on which to fuse biomolecules for several functions. Indeed, the surface of the exosomes has a large N-terminal extra-membrane domain of LAMP-2B, which provides a golden ticket for researchers to connect biomolecules and therapeutic agents [49]. For instance, engineering to express LAMP2B on the surface of the mouse immature dendritic cells (imDCs) was achieved via the fusion of the αv-integrin-specific (iRGD) peptide (CRGDKGPDC), which reduces the immunogenicity and toxicity of the extraneous exosomes. imDC-derived EVs were naturally equipped with the iRGD peptide, improving the tumor-targeting capability [33]. These iRGD exosomes were used to specifically target αvβ3-harboring A549 tumors via delivering the KRAS siRNA in vivo, which resulted in tumor suppression via the knocking down of the KRAS gene [50].

Similarly, the N-terminal of LAMP2B fused with interleukin-3 receptor (IL-3Rα) improved exosome-targeting efficiency at treating chronic myeloid leukemia (CML) [51]. The IL-3Rα-exosomes, derived from CML cells with highly expressed IL-3Rα, were further loaded with breakpoint cluster region (BCR)-ABL siRNA and imatinib. These exosomes were highly accumulated at the tumor site, showed an intense anti-tumor effect and increased survival rate of xenografted mice. Another study used a similar strategy to fuse the N-terminus of LAMP2B with HER2 and efficiently targeted colon cancer [52]. The surface of the exosomes expressing HER2-LAMP2B fusion protein promoted tumor-specific uptake via epidermal growth factor receptor (EGFR)-mediated endocytosis. The incorporated HER2-LAMP2B proteins were expressed on exosomes with 5-fluorouracil (5-FU) and miRNA (target-HER2-LAMP2-GFP) via electroporation and incubation, respectively. These exosomes accumulated well on the tumor site and showed extensive suppression of colon cancer in BALB/c nude mice [52]. Likewise, a lentiviral construct containing LAMP-2B-DARPin the G3 chimeric gene was transduced in HEK293T cells to produce exosomes bearing DARPin G3 [53]. Exosomes were loaded with siRNA for targeted delivery to SKB3 tumor cells. These exosomes could specifically target the SKB3 cells and deliver siRNA to inhibit gene expression [53]. Another study transfected the HEK293T cells with a tLyp-1 (tumor-homing and -penetrating peptide CGNKRTR) LAMP2B plasmid construct. The derived exosomes from these cells were electroporated to be loaded with the synthesized siRNA [54]. These tLyp-1-siRNA exosomes showed enhanced delivery for lung cancer via selectively targeting neuropilin receptors (NRP1 and NRP2) expressed on the tumor tissues. 

In addition to LAMP2B, the transmembrane protein platelet-derived growth factor receptor (PDGFR) is another commonly used membrane display. In a study, the PDGFR transmembrane region GE11 (YHWYGYTPQNVI) was genetically fused using a phage-display vector transfected in the 293T cells to produce the exosome with GE11 [55]. These exosomes showed low mitogenic activity and a high affinity for EGFR-overexpressing cancer cells; thus, they were demonstrated as part of a tailor-made delivery system for EGFR-targeted therapy. The GE11 exosomes were loaded with anti-tumor nucleic acid inhibitor miRNA let-7 and consistently inhibited mouse tumor growth [55]. Similarly, two antibody fragments (αCD3 UCHT1 and scFv fragments of αEGFR cetuximab) were genetically introduced on the exosome surface [56]. This work demonstrated how the cross-linking of EGFR-expressing breast cancer cells and T-cells effectively promoted anti-tumor immunity. Furthermore, a study combined immune checkpoint blockade and oncolytic virotherapy in single-particle nanovesicles with programmed cell death protein 1 (PD1) to create bioengineered cell membrane nanovesicles (PD1-BCMNs) [57]. The PD1-BCMN nanovesicles were harbored with oncolytic adenovirus (OA). PD1-BCMN nanovesicles specifically delivered the OA to immunologically cold tumor tissue, turning it into an immunologically hot tumor. This led to the presentation of more targets for enhanced delivery and showed a strong anti-tumor immune response via effectively activating the tumor-infiltrating T-cells [57].

### 3.2. EV Signature Protein Fusion

Exosomes highly express signature proteins, such as CD9, CD63, and CD81, that can be fused with the targeting molecules [58]. In particular, Ran et al., 2020, fused myostatin propeptide with the second extracellular loop of CD63, which increased exosome serum stability and its delivery efficacy in MDX mice [59]. The exosome surface presents a high-density lipoprotein (HDL), ApoA-1, which binds to the scavenger receptor class B type 1 (SR-B1) abundantly located on hepatocellular carcinoma cells. Liang et al., 2018, genetically introduced Apo-A1 in 293T cells, subsequently inserting the extracellular loop of CD63 on the surface of exosomes [60]. The exosomes derived from the ApoA-1-overexpressing donor cells were primed with miR-26a via electroporation. These engineered exosomes selectively bound the HepG2 cells via SR-B1 and were captured via receptor-mediated endocytosis, leading to the release of miRNA in HepG2 cells, reducing cell migration and proliferation [60]. Another study fused CD63 with the ovalbumin antigen (OVA-Ag), transfecting encoding plasmid DNA into parent cells to produce OVA-carrying exosomes [61]. Vaccinating mice with these exosomes showed a strong Ag-specific CD8+ T-cell response repressing tumor growth [62]. Exosome secretion and uptake were visualized by fusing the extracellular loop of CD63 with the fluorescent protein pHluorin. 

Similarly, functionally customized exosomes were made through genetic modification to accommodate actively integrated membrane proteins or soluble protein cargos (GIFTed-Exos) [63]. The remarkable properties of GIFTed-Exos that stimulate T-cells were obtained by genetically combining the glucocorticoid-induced tumor necrosis factor receptor family-related ligand (GITRL) with exosome-associated tetraspanin CD9 and transmembrane protein CD70. The feasibility of genetically linking the fluorescent protein mCherry to a membrane protein fusion facilitated the release of apoptin-inducing proteins, of apoptin, and of antioxidant enzymes through light-stimulated delivery. Therefore, a vast array of proteins can be delivered to the target cells through GIFTed-Exos [63]. Surface-displayed antigens on exosomes have been used as in anti-cancer vaccines. A study fused CD63 with OVA-Ag to produce OVA exosomes, improving DNA vaccine immunogenicity and preventing mouse tumor growth [61]. The PS-containing exosome membrane has localized C1C2 domains of lactadherin that can be used for anchoring recombinant proteins to promote the increase in cytokine levels for immunogenicity and therapeutic efficacy in TME. This approach successfully suppressed murine lymphoma via exosome-targeting tumor antigen vaccines in the in vivo model [64]. The exosomal protein vesicular stomatitis infection glycoproteins (VSVGs) are also available on exosome surfaces [65]. The VSVG contains a cytoplasmic region and a transmembrane region. Upon replacing the extracellular and cytoplasmic regions of VSVG with the other proteins (red fluorescent proteins (RFP), luciferase, or green fluorescent proteins (GFP)), a permissible enrichment of surface proteins was created without altering the transmembrane space and symbolic peptide. The surface presentation using VSVG fusion proteins on the exosome promoted the target cell uptake of engineered exosomes via the enhanced protein surface display [66].

### 3.3. Genetic Engineering Cancer Cell-Derived EVs

In a proof-of-concept study, genetic manipulation of exosomes was achieved via transfection of lung adenocarcinoma cells, SK-LU-1 cells. The isolated exosomes were shown to promote the transmission of TAMs to the M1 active profiles [67]. Likewise, human pancreatic cancer cell (Panc-1 cell)-derived exosomes co-transfected with HA-PEI/HA-PEG NP transmissive miR-155 and miR-125b2 plasmid DNA had synergistic roles in converting M2-like macrophages into M1-like ones [68]. Additionally, genetically engineered human epidermal growth factor and anti-HER2 antibodies have been used as targeting moieties for targeting MDA-MB-468 tumor xenografts [69]. In contrast, the biofunctionalized liposome-like nanovesicles (BLNs) synthesized via attaching the hEGF covalently to the artificial liposomes held outstanding targeting capabilities and high biological functionalities. In addition, doxorubicin (DOX)-conjugated BLNs exhibited significantly higher anti-tumor therapeutic outcomes than the liposome doxorubicin (Doxil), a clinically approved chemotherapy, did [69]. Given the ease of manufacturing and excellent targeting capabilities, BLNs are a promising alternative for immune liposomes and proteoliposomes. In another recent study, α-lactalbumin (α-LA)-engineered breast cancer cell-derived exosomes were loaded with immunogenic cell death (ICD) inducers Hiltonol (a Toll-like receptor agonist) and human neutrophil elastase (ELANE) to form an in situ DC vaccine (HELA-Exos) [70]. Exposure to Hiltonol and tumor antigens adequately stimulated the induction of immunogenic cell death (ICD) in cancer cells by HELA-Exo. This process activated type-one conventional DCs (cDC1s) locally and triggered a robust CD8+ T-cell response against tumor cells. Consequently, this immune response effectively suppressed poorly immunogenic triple-negative breast cancer (TNBC) in both a xenograft mouse model and patient-derived tumor organoids [70]. 

Similarly, the NIH 3T3 cell lines were engineered to express IL-15/IL-15R𝛼 on the cell membrane. Nanovesicles derived through extrusion overexpressing the IL-15/IL-15R𝛼 (IL-15/IL-15R𝛼-NVs) complex boosted the proliferation, activation, and survival of tumor-infiltrated T-cells via the trans presentation of IL-15 to T-cells and the eliminated tumor. These nanovesicles also promote the activation and proliferation of tumor-specific CD8+ T-cells and TRM cells, effectively inhibiting melanoma growth in mice and increasing the survival rate of mice. An IL-15/IL-15R𝛼-nanovesicle complex was used as cargo to carry the PD-1/PD-L1 inhibitor to a T-cell. The PD-1/PD-L1-inhibitor combined with the IL-15/IL-15R𝛼-nanovesicle complex significantly enhanced the anti-tumor responses [71]. Similarly, Shi et al., 2020, expressed an anti-CD3-anti-HER2 bispecific scFv antibody in Expi293 cells and derived a synthetic multivalent antibody-retargeted exosome (SMART-Exo) to control cellular immunity. The derived SMART-Exo targeted breast cancer-associated HER2 receptors and the T-cell CD3 (Figure 2a). By activating cytotoxic T-cells and redirecting towards attacking HER2-expressing breast cancer cells, the SMART-Exo exhibited specific anti-tumor activity and high potency [41].

CXCR4 is a chemokine receptor that regulates T-cell migration. In a study, MSCs were infected with the PGMLV-PA6-containing virus expressing CXCR4 protein and GFP [72]. The derived exosomes exhibited high CXCR4 expression as a targeted gene–drug delivery system. In addition, the surviving gene (si-survivin) was loaded via electrotransformation to target the tumor site and inhibit growth. RNAi was loaded in the gene–drug delivery system (CXCR4high Exo/si-Survivin) to target lung and gastric cancer. [72]. In another study, murine melanoma B16BL6 cells were transfected with a fusion of streptavidin–cadherin (SAV-LA) protein-encoding plasmid yielding SAV-LA-expressing exosomes (SAV-Exo). It was modified by combining biotinylated CpG (5′-C-phosphate-G-3′) DNA-producing CpG-DNA-modified exosomes (CpG-SAV-Exo). The CpG-SAV-Exo exhibited efficient delivery of the exosome with CpG DNA which promoted the activation of murine dendritic DC2.4 cells in culture and enhanced their the tumor antigen presentation capacity(Figure 2b) [73].

**Figure 2 cancers-15-02838-f002:**
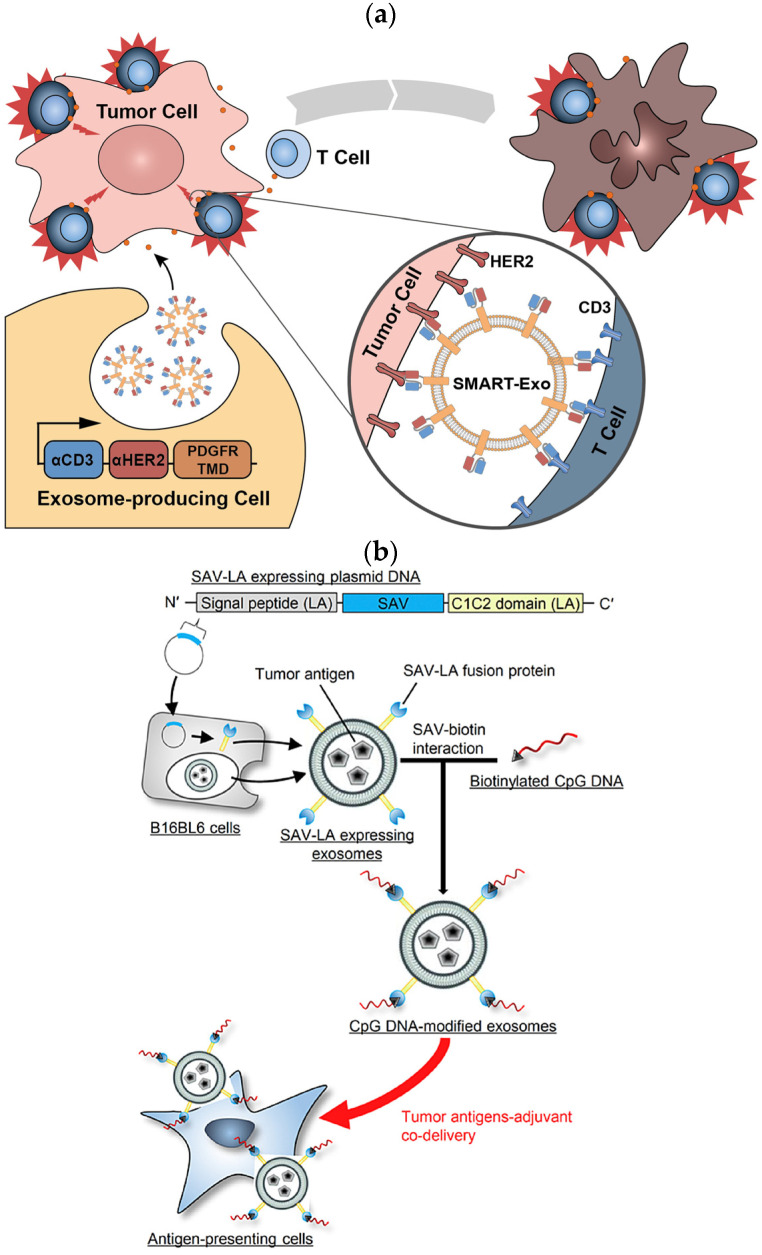
(**a**) Schematic representation of the surface-expressed SMART exosomes derived from the Anti-CD3-anti-HER2-engineered Expi293 cells exhibiting a bispecific scFv antibody, targeting breast cancer-associated HER2 receptors and the T-cell CD3. (**b**) Preparation of SAV-LA-expressing exosomes (SAV-exo). B16BL6 cells were transfected with plasmid DNA encoding streptavidin (SAV) fusion with lactahedrin (LA). SAV-exo were collected from the culture supernant of B16BL6 cells. Further, CpG-SAV-exo were prepared by mixing SAV-exo and biotinylated CpG DNA for enhanced tumor antigen presentation (red arrow). Adapted with permission from Shi et al., 2020 and Morishita et al., 2016 [41,73].

In another study, MSCs were transfected with the pEGFP-C1-GFE1-LAMP2B plasmid. The functional exosome (GExoI) was decorated with pulmonary targeting peptide GFE1 on the membrane surface. These exosomes were loaded with the PI3Kγ inhibitor (IPI549) to suppress melanoma lung metastasis. In a postoperative mouse model, the accumulation of intravenously injected GExoI in the lungs released IPI549 to block G-MDSC recruitment by interfering with CXCLs/CXCR2/PI3Kγ signaling. The increased percentages of CD4+ and CD8+ T-cells in the lungs inhibited metastasis and immunostimulation of the TME [74]. In another study, non-small cell lung cancer (NSCLC) cell lines were overexpressed with PD-1. The biomimetic nanovesicles derived from the NSCLC cell lines exhibited PD-1 (P-NV) and efficiently targeted the NSCLC cells. Further loading P-NVs with doxorubicin (DOX) and 2-deoxy-D-glucose (2-DG) efficiently shrank autochthonous and allografted lung cancers in a mouse model. This P-NV-loaded DOX effectively caused cytotoxicity and activated the anti-tumor immune function of infiltrating T-cells [75]. In another study, U937 monoblastic cells were engineered with the anti-PSMA peptide (WQPDTAHHWAT) to generate exosome mimetics (Ems). These EMs presented an anti-PSMA peptide that targeted advanced prostate cancer (PC). In addition to anti-PSMA peptides, these nanosized EMs displayed monocyte proteins and exosomal markers on the surface. The cellular internalization of the anti-PSMA-EMs was increased in PSMA-positive PC cell lines (LNCaP and C4-2B) and enhanced tumor targeting in solid C4-2B tumors [76].

## 4. Chemical Conjugation Strategies for EV Surface Engineering

The lipid bilayer of the EV surface primarily consists of phospholipids and transmembrane proteins that can be functionalized via chemical conjugation [77]. Chemical conjugation is a direct surface engineering method that uses chemical reagents to add functional moieties. These moieties may include peptides, antibodies, fluorescent tags, and other moieties capable of immune modulation in the TME [78]. Chemical conjugation can include covalent and non-covalent modifications. We discuss the chemical conjugation approach recently used for EV surface modification. 

### 4.1. Covalent Modifications

The covalent modification involves the direct conjugation of ligands to the EVssurface. Functional moieties such as those of carboxylic acid, amines, and sulfhydryl on the EV surface enable covalent conjugation reactions [79]. The widely used strategies for covalent conjugation include the coupling of thiol–maleimide, EDC–NHS coupling, bioorthogonal click chemistry, and amidation chemistry. The thiol–maleimide coupling reaction involves adding maleimide to the sulfhydryl groups on the surface of EVs [80]. Geng et al., 2023, used thiol–maleimide coupling or conjugation of cyclic arginine-glycine-aspartic acid-tyrosine-cysteine (cRGDyC), a ligand selectively targeting integrins (αvβ3), which is overexpressed in glioblastoma cells, to secrete EVs via a two-step process [81]. This conjugation did not impact the structural integrity of EVs and had a higher tumor-targeting capacity. A study by Di et al., 2019, demonstrated the addition of maleimide moieties to engineer EVs. The maleimide approach holds great promise since it can be readily replaced with other functional ligands capable of modulating the TME [82]. Similarly, Zhu et al., 2019, directly modified the EV surface by conjugating cRGDyK and showed enhanced tumor targeting [83]. Tian et al., 2021, also employed this strategy for conjugating T-cell-derived exosomes with anti-EFGR nanobodies [84]. Briefly, the anti-VEGF was functionalized with maleimide by cyclic arginine-glycine-aspartic acid-tyrosine-proline(cRGDyK/cL), a peptide cleavable by metalloproteinase enzymes. Further, the functionalized anti-VEGF nanobodies were cross-linked with T-cell-derived EVs via thiol–maleimide coupling. The yielded modified EVs (rEXS–cL–aV) were capable of suppressing neovascularization. Likewise, Zhou et al. developed a pancreatic ductal adenocarcinoma (PDAC)-targeting exosome-based bio-platform for improved tumor targeting efficacy [85]. Exosomes derived from bone marrow MSCs (BM-MSCs) were modified using oxaliplatin (OXA). They functionalized OXA with N-(2-Aminoethyl) maleimide to synthesize OXA-maleimide and integrated it into BM-MSC-secreted EVs (loaded with siRNA) via thiol–maleimide coupling. Recently, Jung et al., 2022, conjugated dendritic cells with anticytotoxic T-lymphocyte-associated protein-4 antibodies (aCTLA-4) using thiol–maleimide coupling to enhance the therapeutic efficacy on the tumor [86]. This work demonstrates the implications of immune checkpoint blockade (ICB) therapy in cancer treatment and elaborates on its role in optimization of delivery systems for enhanced therapeutic outcomes with minimal side effects.

EDC–NHS coupling could be used for conjugating peptides, proteins, antibodies, and so on to the EVs surface for tumor therapy. The EDC–NHS coupling reaction involves using 1-ethyl-3-(3-dimethylamino propyl)-carbodiimide, which mediates the reaction between carboxyl groups and primary amines by forming amine-reactive NHS esters [87]. Hosseini et al., 2022, demonstrated the use of AS1411 aptamer-functionalized exosomes for the targeted delivery of doxorubicin. Here, EDC–NHS coupling was used to convert the AS1411 aptamer into amine-reactive NHS esters for exosome conjugation [88]. By employing this strategy, Xu et al., 2021, conjugated a polyarginine peptide (R9) to develop a peptide-equipped exosome [89]. They coupled the R9 peptide with the carboxyl group on the surface of HepG2 cell-derived exosomes via an EDC–NHS-mediated amide reaction. Likewise, Choi et al., 2019, reported that NHS coupling chemistry could be used to conjugate PEG to the surface of EVs, leading to an increased tumor-targeting ability [90].

Among all the conventional methods, bio-orthogonal click chemistry is highly efficient and biocompatible, and can be effectively employed for the surface engineering of EVs. Click chemistry (azide–alkyne cycloaddition) is a reaction between an alkyne and an azide, involving the catalysis of a triazole linkage by copper [91]. Smyth et al., 2014, created a novel surface functionalization method for EVs via click chemistry. Briefly, the conjugation of azide-fluor-545 to an alkyne group modified exosomes [92]. The study suggested that azide functionalization using click chemistry had no effect on the structure and functions of exosomes and was found to be efficient for a wide range of applications. Jia et al., 2018, conjugated arginine–glycine–glutamic acid (RGE) and fluorescein isothiocyanate (FITC) via click chemistry to target glioma cells [93]. Exosomes were produced by loading curcumin (Cur) and superparamagnetic iron oxide nanoparticles (SPIONs); then, using a cycloaddition of sulfonyl azide, the exosome membrane was coupled to a neuropilin-1-targeted peptide (RGERPPR, RGE), and FITC to create glioma-targeting exosomes with imaging and therapeutic properties. One caveat is that using copper in click-chemistry could cause cytotoxicity; thus, a copper-free reaction, namely a strain-promoted azide–alkyne cycloaddition (SPAAC), has also been developed. Azide groups efficiently undergo Huisgen-type cycloaddition with strained cyclo-octyne groups, resulting in a stable triazole ring [79]. In a two-step process, cyclo-peptides (Arg-Gly-Asp-DTyr-Lys) were conjugated to the surface of exosomes. The first step involved the amino groups on the exosomal proteins reacting with the dibenzocyclooctyne (DBCO) Sulfo-NHS esters. The second step involved contact of the azide-functionalized cyclo-peptides with DBCO groups. These conjugated exosomes showed more efficient tumor-targeting [79]. Nie et al., 2020, conjugated azide-modified exosomes with the DBCO-modified antibodies of CD47 and signaled regulatory protein alpha (SIRPa) through a pH-sensitive linker [94]. The conjugated exosomes could effectively target tumors via anti-CD47 antibody and CD47 receptor interaction. In acidic TME, benzoic amine bonds cleave and release SIRPα and anti-CD47, enhancing tumor cell phagocytic elimination. Employing a similar strategy, Wang et al., 2015, combined metabolic engineering with bioorthogonal click chemistry to engineer EV surfaces [95]. This study showed that click chemistry, i.e., azide-integrated exosomes, can be conjugated to various functional moieties and applied for a wide range of applications, including the immune modulation of TMEs. In conclusion, either copper-catalyzed or copper-free click chemistry could be used to conjugate exosomes with functional ligands capable of tumor modulation.

### 4.2. Non-Covalent Modifications

Unlike covalent modifications, non-covalent modifications correspond to EV conjugation via weak interactions, including electrostatic, hydrophobic, or ligand-receptor interactions by nature [79]. Non-covalent modifications of EVs are relatively easy to perform. EV surface modifications via electrostatic interactions are achieved by adding functional moieties that can impart the positive charge to the negatively charged surface and enhance EV targeting toward biological membranes [96]. Mizuta et al., 2019, modified the surface of exosomes via a non-covalent and hydrophobic interaction [97]. They conjugated magnetic nano gel comprising cholesterol-bearing pullulan (CHP) and iron oxide nanoparticles to the exosome surface and demonstrated a high efficacy of cell targeting. Likewise, Tamura et al., 2017, used cationized pullulan to further increase therapeutic efficacy with a high affinity towards the asialoglycoprotein receptors expressed on liver cells [98]. In another study, Nakase et al., 2015, used a simple method for the surface modification of exosomes; they used a commercially available cationic lipid formulation, Lipofectamine LTX, and pH-sensitive fusogenic peptide GALA [99]. These modifications led to the efficient induction of cytotoxicity in the cells. Koh et al., 2017, modified the EV surface with an immune checkpoint blockade that antagonized the interaction between CD47 and SIRPα, leading to the phagocytic elimination of tumor cells [100]. Similarly, Zhan et al., 2020, anchored the cationic lipid-sensitive endosmotic peptide, L17E (lipid-sensitive endosmotic), to the exosome membrane via electrostatic interaction, enhancing tumor targeting [101]. Although cationic modification is efficient, its application is hazardous as the endocytosis of cations could lead to lysosome degradation [96]. 

Among others, the non-covalent modifications applied in EV modification are ligand-receptor interactions. This method involves the conjugation of ligands to increase EV target specificity. Qi et al., 2016, reported a notable illustration of modified reticulocyte (RTC)-derived exosomes with superparamagnetic nanoparticles (SPMN). They conjugated SPMN with transferrin (Tf) and attached Tf-conjugated SPMN to the RTC-derived exosomes with TF receptors on their surface through the Tf and Tf-receptor interaction [102]. These modified exosomes were capable of robust tumor targeting. Moreover, dual ligand conjugations have also demonstrated high anti-tumor responsivity. Wang et al., 2017, conjugated EVs with a biotin and avidin complex to improve the tumor-targeting ability; the avidin ligand bound to its lecithin receptor that was overexpressed in cancer cells [103]. Similarly, Liu et al., 2019, surface-engineered EV with lipidomimetic compound-modified hyaluronic acid derivatives carrying octadecyl tails (lipHA [104]. These lipHA-hEVs possessed high tumor targeting due to binding to CD44, which is overexpressed in multi-drug-resistant cancer. Maguire et al., 2012, incorporated biotinylated magnetic nanoparticles onto EV surfaces [105]. Wang et al., 2021, used polydopamine (PDA) for the non-covalent modification of exosome surfaces. PDA exosomes could be further modified with functional molecules such as PEG, fluorophores, antibodies, or other functional moieties via secondary reactions, including thiol coupling, click chemistry, and other covalent conjugations [106].

EV membranes primarily comprise amphiphilic substances such as phospholipids, cholesterol, and glycolipids, making them conducive to the integration of hydrophobic compounds through hydrophobic interactions. Hydrophobic interactions involve the spontaneous integration of hydrophobic moieties on the EV surface. Aminoethyl anisamide-polyethylene glycol (AA-PEG) was conjugated to vectorized exosomes via hydrophobic interaction. AA-PEG-vectorized exosomes showed high tumor targeting, given their interaction with the sigma receptor that is overexpressed in lung cancer cells. These engineered exosomes (EXO) were attached with mannose, a natural sugar with anti-tumor properties [90]. The EV surfaces were modified with PEG via incorporating 1,2-Distearoyl-Sn-Glycero-3-Phosphoethanolamine (DSPE) onto the EV membrane and were further functionalized with mannose to obtain EXO-PEG-man, which could efficiently deliver immunomodulators to lymph nodes. The diphosphine ligand 1,2-bis(dimethyl-phosphine)ethane (DMPE) has been widely used for the PEGlyation of EVs to increase circulation times. Employing this strategy, Kooijmans et al., 2016, decorated EVs with targeting ligands conjugated to DMPE-PEG [107]. They conjugated the nanobodies specific for EGFR with the DMPE-PEG derivative to functionalize EVs with these nanobody–PEG micelles. Modified EVs have enhanced cell specificity and prolonged circulation times. Thus, using ligand-conjugated PEG-phospholipid derivatives, EVs could be decorated with specific ligands capable of immune modulation in the TME. In another study by Antes et al., 2018, exosomes were attached with DMPE-PEG embedded into the membrane as an anchor for conjugating targeting moieties to functionalize EVs [108]. Likewise, Jiang et al., 2021, investigated the lipid-mediated post-insertion strategy for surface engineering mammalian- and bacteria-derived EVs. The study reported the method as being comparatively more rapid than other methods [109]. Thus, cholesterol-mediated insertion could be used for the surface engineering of EVs.

In terms of folate-mediated EV modifications, folic acid (FA) receptor is a glycoprotein receptor that is overexpressed in cancer cells [110]. Exosomes have been conjugated with FA for enhanced tumor targeting. Feng et al., 2021, used folate–PEG to conjugate exosomes for tumor immune modulation [111]. A combined genetic engineering and hydrophobic insertion strategy was adopted to engineer the exosome expressing human hyaluronidase (PH20) to obtain Exo-PH20-FA. Further, Zheng et al., 2019, decorated exosomes with FA for the cytosolic delivery of small interfering RNA (siRNA) [112]. FA-exosome-siRNA enhanced cancer suppression compared to FA-siRNA only. Li et al., 2018, also employed this strategy to modify exosomes [113]. They used arrow-tail RNA to display FA ligands on ginger-derived exosomes for effective tumor-targeted siRNA delivery. Yu et al., 2019, surface-engineered exosomes with FA to produce FA-vectorized exosomes [114]. These exosomes, loaded with erastin, were conjugated with DSPE-PEG-FA and induced ferroptosis (lipid peroxide-driven cell death) in TNBC cells. Zhu et al., 2017, modified the EV membrane via the phospholipid substitution strategy [115]. They functionalized microvesicles (MV) with biotin and folate (BFMV) and loaded them with Bcl-2 siRNA and paclitaxel through electroporation. BFMVs have shown high levels of tumor targeting due to folate, which has a synergistic anti-tumor effect. Likewise, Zhang et al., 2017, developed magnetic and FA-modified MVs via a donor-assisted membrane modification strategy; modified MVs exhibited enhanced anti-tumor efficacy [116]. 

In terms of liposome-mediated surface engineering, liposomes are small artificial spherical vesicles used for drug delivery owing to their size and amphiphilic nature [117]. Liposome fusion could be used to modify the surface of EVs. A study by Lee et al., 2016, demonstrated that liposome-based cellular engineering could prepare clickable EVs; they incorporated various functional moieties through biorthogonal chemistry [118]. Similarly, Piffoux et al., 2018, developed a hybrid EV-fusing liposome on the EV membrane and used PEG to mediate the fusion between liposomes on the EV surface [119]. This PEG-mediated fusion enabled the development of a hybrid EV that was tunable without alterations to its inner cargo and activity. 

With regard to aptamer-based surface engineering, aptamers are short artificial oligonucleotides and can be attached to EV surfaces, given their high binding ability and specificity. Both covalent and non-covalent coupling methods are employed for aptamer conjugation. Hosseini et al., 2022, used the AS1411 aptamer to functionalize exosomes for the targeted delivery of doxorubicin [88]. The AS1411 aptamers were converted into amine-reactive NHS esters to conjugate with the exosome using te EDC–NHS coupling method. Similarly, the nucleolin-targeting aptamer AS1411 was covalently conjugated to cholesterol–PEG and anchored onto the surface of MSC-derived EVs, yielding higher tumor-targeting capacity [120]. Likewise, Bagheri et al., 2020, conjugated MSC-derived exosomes with carboxylate-modified MUC1 aptamer (5TR1) [121]. The MUC1 aptamer was coupled to amine groups on MSC-EV surfaces through EDC–NHS chemistry. The same research group also employed the surface functionalization of EVs using a aptamer via EDC–NHS coupling [122]. They conjugated the LJM-3064 aptamer to the surface amines of EVs for enhanced targeting. Another strategy for aptamer-based modification is through a thiol–maleimide addition. Han et al., 2021, modified HEK293T cell-derived exosomes by conjugating aptamer onto the surface of exosomes via thiol–maleimide conjugation chemistry for inhibiting metastatic prostate cancer [123].

## 5. Physical Methods

Physical surface engineering methods include electroporation, sonication, extrusion, and freeze–thaw. These methods temporarily disrupt the lipid constructs of membranes and permeabilize the EV membrane for cargo loading and surface functionalization. Freeze–thawing, electroporation, and extrusion are common physical methods that can be used for EV surface permeabilization and modifications. These techniques create surface pores to allow modifying agents to pass through or bind to exposed membrane lipids or proteins. The freeze–thaw method disrupts the membrane via the formation of ice crystals. Sato et al., 2016, utilized this strategy and developed hybrid exosome—liposome-fused vesicles that combine the advantage of natural targeting and signaling properties of exosomes with the drug-delivery capabilities of liposomes. The study demonstrates that hybrid exosomes efficiently deliver therapeutic molecules by targeting HeLa cells. Likewise, Cheng et al., 2019, used freeze–thaw techniques to incorporate nuclear localization signal peptides on the EV surface by shaking the peptide and EVs together in an ice bath for four hours. The resulting modified EVs exhibited significantly improved therapeutic effects on inhibiting tumor growth [124]. However, repeated temperature changes during freeze–thaw cycles may lead to the denaturation of transmembrane proteins and affect the stability of resultant EVs. Alternatively, extrusion-based membrane fusion has been used to overcome the denaturation issue. 

Electroporation is a technique that uses an electric field to temporarily disrupt the cell membrane, allowing the introduction of exogenous materials into the membrane of EVs to enhance their tumor targeting and therapeutic capabilities. Although it is a method of choice for loading hydrophilic therapeutic cargo (i.e., DNA and RNA) onto EVs, its implications in EV surface engineering remain limited due to non-uniformity in terms of incorporation and orientation. On the other hand, microfluidic isolation and molecular analysis of EVs have seen significant progress. However, microfluidic engineering of exosomes has just emerged recently, and a few reports have employed microfluidic technology for engineering exosomes. Their full potential and capability have yet to be thoroughly explored. Due to their inherent customizability, automation, scalability, and capacity for highly efficient mass transport, microfluidic systems can overcome numerous limitations of benchtop systems. Microfluidic lab-on-chip technology has demonstrated the potential of these advantages [125]. Dendritic cell-derived immunogenic exosomes possess an intrinsic payload of MHC class I and II molecules and other co-stimulatory molecules that aid in mediating immune responses. Akagi et al., 2015, used a microfluidic cell culture system to engineer the surface of cultured immunogenic exosomes (MHC I-positive) with tumor antigenic peptides. These functional exosomes exhibited enhanced cellular uptake by antigen-presenting cells compared to non-engineered exosomes [126].

Extrusion is a common physical modification technique for transferring EV membranes onto different nanoparticles. For instance, EV membrane-cloaked gold nanoparticles were synthesized by Deun et al., 2020, to limit the uptake of gold nanoparticles by macrophages [127]. A similar EV membrane biomimicking approach has also been adopted for other nanoparticles and polymeric nanoparticles; however, their potential application in tumor immunotherapy remains unexplored [128,129]. 

Coincubation is a common method with which to introduce targeting peptides onto extracellular vesicle (EV) surfaces. This process involves incubating EVs with the targeting peptide of interest, which can bind to specific receptors on the target cells. Kooijmans et al., 2018, demonstrated the potential of EVs by decorating EVs with EGa1-C1C2 for tumor targeting [130]. Here, lactadherin (C1C2) was fused with EGFR nanobodies (EGa1), and EGa1-C1C2 was incorporated in the EV membranes via coincubation. This conjugation conferred the EVs with high tumor targeting ability without influencing the structure of the EV. Thus, recombinant C1C2-fused proteins could be used with therapeutic proteins and other ligands for effective tumor targeting. Pham et al., 2021, reported a simple enzymatic method using protein-ligating enzymes sortase A and OaAEP1 to conjugate EVs and peptides by forming permanent covalent bonds [44]. The surface functionalization of EVs with target-specific peptides and nanobodies can improve the transport of therapeutic compounds to cancer cells expressing the associated ligands, enhancing treatment efficacy, and reducing adverse effects.

## 6. Pre-Clinical and Clinical Utility of Surface-Engineered EVs

Pre-clinical studies investigating surface-modified EVs for cancer immunotherapy were categorically outlined in the above sections. Table 1 provides a summary of these strategies. In addition, Table 2 broadly summarizes the preclinical studies reporting the efficacy of surface-modified EVs. Given these promising outcomes from pre-clinical studies, there is expected to be a drastic surge in clinical trials in the near future. The advancement of EV-based therapeutics and its applications is evident from the number of ongoing clinical trials investigating EV potential—a survey of ClinicaTrials.gov was conducted to find clinical trials investigating surface-modified EVs in cancer. We searched clinical trials with the keywords ‘extracellular vesicles’ or ‘exosomes in cancer interventions’. Most clinical trials were focused on the molecular profiling of circulating EVs in liquid biopsies to assess their role in disease as potential biomarkers for diagnostic, prognostic, and predictive values. Rezaie et al., 2022, comprehensively reported exosome applications in clinical trials [131]. 

While surface-engineered EVs remain to be broadly evaluated in clinical trials, we noted two studies using surface-modified EVs for cancer therapy. One evaluated dendritic cell-derived exosomes (dexosomes or Dex) in advanced NSCLC patients (NCT01159288). While the primary endpoint of this trial was not reached, the phase-I and phase-II studies confirmed that Dex boosts the NK cells’ antitumor immunity in patients [159]. Dex was initially developed as an alternative approach to cancer vaccinations and was surface-modified for immunomodulatory functions [163]. The surface molecular composition of Dex allows effective targeting and docking to recipient cells with robust immunostimulatory functionality [164]. Another phase-I study used ascite-derived exosomes (Aex) in combination with granulocyte–macrophage colony-stimulating factor (GM-CSF) in the immunotherapy of colorectal cancer [162]. The study reported that the Aex and GM-CSF combination is safe and well-tolerated and induces beneficial antitumor cytotoxic T lymphocyte responses. 

Although EVs have been extensively researched, their clinical translation has been deterred by many factors. The lack of standardized large-scale EV production limits clinical applications [165]. Conventional EV isolation methods are challenging as they affect the physiochemical properties of EVs.. However, some drawbacks are mitigated using novel isolation techniques based on membrane separation and microfluidics [166]. The surface heterogeneity of EVs hampers its characterization, which limits the interpretation of the biodistribution of EVs in the recipient’s biological system [167]. In addition, storage and preservation conditions for EVs also play a vital role in structural integrity and functional efficacy [168]. The challenges in EV production, isolation, characterization, and surface modifications must be addressed to effectively translate surface-engineered EVs from lab applications to clinical applications.

## 7. Conclusions and Future Directions

The emergence of nanotechnology in medical applications has introduced a new era of drug delivery. Due to their high biocompatibility, EVs have shown great promise as therapeutic carriers and in clinical applications. However, there are associated risks in using exosomes, such as potential immunosuppression and reversion to tumorigenesis. As a result, there is a constant pursuit toward achieving safe and effective exosome-based formulations, leading to the external modification of exosomes as a viable approach. EV surface modification using biological, chemical, and physical strategies offers potential solutions to these challenges, improving the effectiveness of EV-based cancer immunotherapies (Table 1). 

Genetic modifications enable the control of cells’ inherent EV production process to generate EVs with specific immunostimulatory or immunosuppressive properties. The incorporation of tumor-associated antigens or cytokines and the expression of chimeric antigen receptors (CARs) or T-cell receptors (TCRs) on EVs are some of the promising strategies for improving the specificity and effectiveness of EV-based immunotherapies. These modifications can enhance EV targeting, stability, and immunostimulatory properties to improve cancer therapeutic outcomes. Chemical modifications of EVs have also shown strong potential for improving their targeting specificity and immunostimulatory properties via attached ligands, antibodies, or peptides onto the EV surface, thus redirecting EVs toward specific immune cells or increasing their specificity to cancer cells. The immunomodulatory properties of EVs have also been achieved by adding adjuvants, such as Toll-like receptor agonists, enhancing EVs’ ability to activate immune cells and stimulate an anti-tumor response. Alternatively, physical modifications, such as size reduction and surface coating with polymers, can enhance EV stability and circulation time, improving their ability to reach and interact with immune cells. 

The new developments in recent years represent an exciting frontier in EV biology and cancer immunotherapy research. There is much to be learned about the optimal approaches for EV modification. However, many obstacles remain to overcome; one is the production of exosomes on a large scale for clinical purposes with more handy EV purification approaches. Rapid EV purification approaches will be an encouraging development in the EV research domain. Any new methods that are yet to come will have to pass through, the scrutiny phase before being adapted for wide application in the field. Additionally, the question of which cell type is optimal for exosome derivation remains unanswered. EVs as delivery vehicles are highly attractive and promising and could be the potential solution to the ongoing quest for clinically viable nanoplatforms. Therefore, conducting more systematic in vivo studies will be crucial for establishing the efficacy and toxicity of exosomes and bringing EVs towards exciting advancement and bringing them closer to clinical implementation.

## Figures and Tables

**Figure 1 cancers-15-02838-f001:**
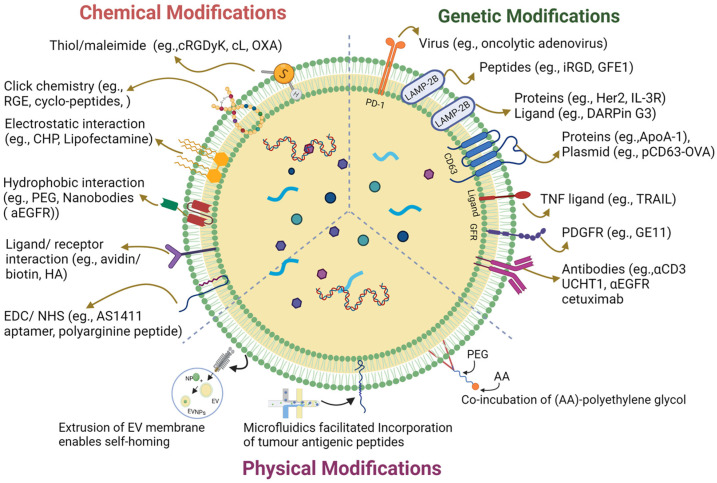
A graphical illustration of extracellular vesicle surface engineering strategies.

**Table 1 cancers-15-02838-t001:** Representative examples of different strategies for EV surface engineering and their implications in cancer immunotherapy.

Incorporation Mechanism/Factors	Modification Strategy	Cancer Type and Targets	References
Genetic Engineering of EV Surface			
LAMP-2B	iRGD	Breast cancer, lung cancer	Tian et al., 2014 [33], Zhou et al., 2019 [50]
	GFE1	Suppression of melanoma lung metastasis	Han et al., 2022 [74]
	Her2	Colon cancer	Liang et al., 2020 [52]
	IL-3R	Chronic myeloid leukaemia	Bellavia et al., 2017 [51]
	DARPin G3	Breast cancer	Limoni et al., 2019 [53]
	tLyp-1	Lung cancer	Bai et al., 2020 [54]
CD63	ApoA-1	Liver cancer	Liang et al., 2018 [60]
	pCD63-OVA	OVA expressing lymphoma cells	Kanuma et al., 2017 [61]
TNF Ligand	TRAIL	Colon and prostate cancer in the blood preventing metastasis	Mitchell et al., 2014 [45], Wayne et al., 2016 [46]
Antibodies	αCD3 UCHT1, αEGFR cetuximab	Breast cancer	Cheng et al., 2018 [56]
PDFR	GE11	EGFR-expressing breast cancer	Ohno et al., 2013 [55]
PD-1	Oncolytic adeno virus	Liver cancer (murine hepatoma cell line)	Lv et al., 2021 [57]
C1C2 domain	Tumor antigens (PAP & PSA)	Anti-tumor activity on the PAP- and PSA-expressing tumor	Rountree et al., 2011 [64]
Breast cancer-derived exosomes	α-lactalbumin (loaded with ICD)	Breast cancer	Huang et al., 2022 [70]
NIH 3T3 cells derived exosome	IL-15/IL-15R𝛼	Melanoma	Fang et al., 2023 [71]
melanoma B16BL6 cells derived exosome	streptavidin (SAV)-lactadherin (LA)	Melanoma	Morishita et al., 2023 [73]
Expi293 cells, the derived exosomes	Anti-CD3-anti-HER2 bispecific scFv antibody	Breast cancer (breast cancer-associated HER2 receptors and CD3 T-cell)	Shi et al., 2020 [41]
U937 monoblastic cells	anti-PSMA peptide	Prostate cancer cells (LNCaP-derived C4-2B)	Severic et al., 2021 [76]
Chemical Modification of EV Surface			
Thiol/maleimide	cRGDyK	Glioblastoma	Geng et al., 2023 [81]
	cL	Suppressed ocular neovascularization	Tian et al., 2021 [84]
	OXA	Pancreatic cancer	Zhou et al., 2021 [85]
	aCTLA-4	Immune checkpoint inhibitor (CTLA-4)	Jung et al., 2022 [86]
Click chemistry	RGE	Glioma	Jia et al., 2018 [93]
	Cyclo-peptides	Cerebral ischemia	Tian et al., 2018 [132]
	aCD47	CD47-overexpressed cells	Nie et al., 2020 [94]
Electrostatic interaction	Cationized pullulan	Liver cells	Tamura et al., 2017 [98]
	L17E peptide	Tumor-targeting and combination therapy	Zhan et al., 2020 [101]
Hydrophobic Interaction	AA-PEG	Liver cancer cells	Kim et al., 2018 [133]
	Nanobodies (aEGFR)	EGFR-over-expressing cells	Kooijmans et al., 2016 [107]
	Avidin/biotin	Lecithin (overexpressed in cancer cells)	Wang et al., 2017 [103]
Ligand/Receptor interaction	HA	Overcoming multi-drug resistance encountered in chemotherapy	Liu et al., 2019 [104]
	SPION	Enhanced targeting toward hepatoma	Qi et al., 2016 [102]
	AS1411 Aptamer	Colorectal cancer	Hosseini et al., 2022 [88]
EDC/NHS	Polyarginine peptide	Liver cells (modified EVs demonstrated a preferential tropism toward parent cells)	Xu et al., 2021 [89]
	PEG	Enhanced circulation time	Choi et al., 2019 [90]
Physical Modification of EV Surface			
Co incubation	Aminoethylanisamide-polyethylene glycol	Enhanced circulation time	Kim et al., 2018 [133]
Extrusion	4T1 tumor	4T1 EV membrane enables self-homing	Bose et al., 2018 [128]
Microfluidics	Incorporation of tumor antigenic peptides in immunogenic exosomes (MHC I+)	Enhanced cellular uptake of engineered exosomes by antigen-presenting cells	Akagi et al., 2015 [126]
Freeze thawing	RAW264.7 derived exosome-Liposome fusion	HER-2-mediated enhanced uptake by HeLa cells	Sato et al., 2016 [134]

**Table 2 cancers-15-02838-t002:** Pre-clinical and clinical trials using surface-engineered EVs in cancer therapy.

Type of Cancer	Origin	Active Pharmaceutical Ingredient (API)	Surface Modification	Reference
Pre-Clinical Trials				
Glioblastoma	RAW264.7 Macrophage	curcumin	Neurophilin-1 targeted peptide	Jia et al., 2018 [93]
Glioblastoma	Malignant cells	CRISPR/Cas9	TNF-α	Gulei et al., 2019 [135]
Glioblastoma	Embryonic stem cells	Paclitaxel	cRGD	Zhu et al., 2019 [83]
Glioblastoma multiforme	L929 cells	Methotrexate and KLA peptide	LDL and KLA peptide	Ye et al., 2018 [136]
Breast cancer	HEK293T	PH20 hyaluronidase and doxorubicin	Folic acid and PH20 hyaluronidase	Feng et al., 2021 [111]
Breast cancer	HEK293T (Expi293)	Anti-CD3 and antiHer2 antibody	Anti-CD3 and anti-Her2 antibody	Shi et al., 2020 [41]
Breast cancer	BMSCs	Doxorubicin	DARPin	Gomari et al., 2019 [137]
Breast cancer	Blood	Chimeric peptide (ChiP)	Chimeric peptide (ChiP)	Cheng et al., 2019 [124]
Breast cancer	4T1 cells	Sinoporphyrin sodium	Sinoporphyrin sodium	Liu et al., 2019 [138]
Breast, prostate, and colorectal cancer	HEK293T	siSurvivin	Folate, PSMA RNA aptamer and EGFR RNA aptamer	Pi et al., 2018 [139]
Breast cancer	Dendritic cells	Paclitaxel	AS1411 aptamer conjugated to cholesterol–PEG	Wan et al., 2018 [120]
Breast cancer	HEK293	HchrR6 mRNA	LS-ML39-C1– C2-His (EVHB)	Wang et al., 2018 [140]
Breast cancer	HEK293	PH20 hyaluronidase and Doxorubicin	PH20 hyaluronidase	Hong et al., 2018 [141]
Breast cancer	Dendritic cells	miRNA let-7 and siRNA-VEGF	AS1411 aptamer	Wang et al., 2017 [142]
Breast Cancer	Immature dendritic cells	Doxorubicin	AlphaV integrin-specific iRGD peptide	Tian et al., 2014 [33]
Breast cancer	HEK293	miRNA-let-7a	Transmembrane domain of platelet-derived growth factor receptor fused to GE11 peptide	Ohno et al., 2013 [55]
Breast cancer multi-drug resistance	HEK293T	Doxorubicin	Lipidomimetic chain-grafted hyaluronic acid	Liu et al., 2019 [104]
Hypoxic breast cancer tumors	MDA-MB-231	Olaparib	SPIO (superparamagnetic iron oxide) nanoparticles	Jung et al., 2018 [143]
Triple-negative breast cancer	HEK293T	PH20 hyaluronidase	PH20 hyaluronidase	Hong et al., 2019 [144]
Triple-negative breast cancer	Macrophages	Doxorubicin and cholesterol-modified miRNA-159	Disintegrin and metalloproteinase 15 (A15)	Gong et al., 2019 [145]
Colorectal cancer	MSCs	Doxorubicin	MUC1 aptamer	Bagheri et al., 2020 [121]
Colorectal cancer	HEK293T	siSur-A647 and folate	Folic acid	Zheng et al., 2019 [112]
Colorectal cancer	LIM1215 cells	Doxorubicin	A33Ab-US	Li et al., 2018 [146]
Colorectal cancer	THLG-293T & LG-293T	Her2-binding affibody	Her2-binding affibody	Liang et al., 2020 [52]
Colorectal cancer	HEK-293T	SIRPα protein	SIRPα protein	Cho et al., 2018 [147]
Colorectal cancer	CT26-CIITA cells	MHC class II molecule	MHC class II molecule	Fan et al., 2013 [148]
Cervical cancer	THP-1 macrophages	Doxorubicin	RGD, sulfhydryl groups, AuNRs and folic acid	Wang et al., 2018 [149]
Cervical cancer	Macrophages	Doxorubicin	Biotin, streptavidin-modified iron oxide nanoparticles SA-IONPs and Folic acid	Zhang et al., 2017 [116]
Lung cancer with mutated KRAS	Bovine milk	siKRAS	Folic acid	Aqil et al., 2019 [150]
Lung cancer	Malignant cells	CRISPR/Cas9	TNF-α	Gulei et al., 2019 [135]
Non-small cell lung cancer	Human plasma	Imperialine	Integrin α3β1-binding octapeptide cNGQGEQc	Lin et al., 2019 [151]
Non-small cell lung cancer	RAW264.7 Macrophage	Paclitaxel	Aminoethylanisamide–PEG	Kim et al., 2018 [133]
Hepatocellular carcinoma	Blood	Doxorubicin	Superparamagnetic magnetite colloidal nanocrystal clusters	Qi et al., 2016 [102]
Hepatocellular carcinoma ascites	BM dendritic cells	Doxorubicin	Tumor-derived antigens	Wu et al., 2017 [152]
Lymphoma	K562 cells	TRAIL protein	TRAIL protein	Rivoltini et al., 2016 [47]
Melanoma	MSCs	TNF-α	Superparamagnetic iron oxide nanoparticles (SPION)	Zhuang et al., 2020 [153]
Melanoma	B16BL6 cells	Immunostimulatory CpG DNA	Streptavidin–lactadherin	Morishita et al., 2016 [73]
Melanoma	Human umbilical vein endothelial cell	siVEGF	Streptavidin-conjugated quantum dots	Chen et al., 2015 [154]
Pancreatic cancer	MSCs	siKRASG12D and pLKO.1- shKRASG12D	CD47	Kamerkar et al., 2017 [155]
Carcinoma (KB xenograft)	Ginger root	siSurvivin	Folic acid	Li et al., 2018 [113]
Chronic myelogenous leukemia	HEK293T	Imatinib & siBCR-ABL	IL3	Bellavia et al., 2017 [51]
Nasopharyngeal cancer	HUVECs	Anti-miR-BART10-5p andAnti-miR-18a	iRGD	Wang et al., 2020 [156]
Neuroendocrine cancer	HEK293	Verrucarin A & romidepsin	Anti-SSTR2 mAb	Si et al., 2020 [157]
Thyroid cancer	Malignant cells	CRISPR/Cas9	TNF-α	Gulei et al., 2019 [135]
Glioblastoma	HEK293T	Anti-miR-21	T7 peptide	Kim et al., 2020 [158]
Clinical Trials				
Non-small cell lung cancer	Dendritic cells	IFN-γ	MHC class I and II	Besse et al., 2016 [159]
Non-small cell lung cancer	Dendritic cells		MAGE antigens	Morse et al., 2005 [160]
Melanoma	Dendritic cells		MAGE 3 peptides	Escudier et al., 2005 [161]
Colorectal Cancer	Ascites-derived exosomes (Aex)	Granulocyte–macrophage colony-stimulating factor	Arcinoembryonic antigen (CEA)	Dai et al., 2008 [162]

## Data Availability

Not applicable.

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
