# Peer review of "Surface-Engineered Extracellular Vesicles in Cancer Immunotherapy"

_cancers, 2023, doi:10.3390/cancers15102838_

Round 1

Reviewer 1 Report

The article called “Surface-Engineered Extracellular Vesicles in Cancer Immunotherapy” is written in detail, relevant. Just a couple of suggestions for authors:

Line 48. In addition to bioactive molecules, vesicles carry organelles (for example mitochondria doi: 10.3389/fcell.2021.653322).

Line 617. There is a permeabilization method used to modify EVs. It also can be mentioned.

Author Response

Response to Reviewer 1

Point 1. Line 48. In addition to bioactive molecules, vesicles carry organelles (for example mitochondria doi: 10.3389/fcell.2021.653322).

Response: Thank you for providing this input. We have included a brief note on mitochondrial EVs with the suggested citation. We have incorporated the following sentence in the manuscript.

“Apart from bioactive molecules, EVs can also encapsulate various cellular organelles, including the transfer of functional mitochondria to other cells, and promote cell survival and tissue regeneration [3].”

Point 2. Line 617. There is a permeabilization method used to modify EVs. It also can be mentioned.

Response: Thank you, we have incorporated a brief note on permeabilization methods to modify EVs under physical methods (now section 5). The following lines were added to this paragraph.

“Freeze thawing, electroporation, and extrusion are common physical methods for EV surface permeabilization and modifications. These techniques create surface pores to allow modifying agents to pass through or bind to exposed membrane lipids or proteins.”

Reviewer 2 Report

The authors of this review discuss the role of EVs in modulating the tumor microenvironment. The review highlights recent progress in genetic, chemical, and physical engineering strategies employed to modify EV surfaces. Additionally, the authors investigate how surface engineering of EVs has emerged as a promising tool for cancer immunotherapy. While the article is scientifically sound and well-written, there are areas that could be improved, as outlined below.

Major points:

1.      On page 6, lines 202-203: Please elaborate on how TRAIL is successfully transfected into the cells. Is TRAIL naturally incorporated into the exosome through simple transfection without any modifications?

2.      On page 7, lines 225-231, these sentences appear rather confusing and require clarification. The abbreviation for interleukin-3 (IL-3Rα) appears to be incorrect.

3.      On lines 243-246: The function of tLyp-1 (tumor homing & penetrating peptide CGNKRTR) and how it enhances exosome delivery into lung cancer should be elaborated.

4.      It would be advantageous to divide Section 3 into smaller subsections since it contains a substantial amount of information presented in a single section.

5.      Is the subsection '4.3 Physical Methods' appropriate to be included in 'Section 4: Chemical Conjugation Strategies for EV Surface'?

6.      It would be good to create a table summarizing the preclinical and clinical trials of the exosome-based therapy being developed as a cancer immunotherapy treatment.

7.      Please define exosome mimetics (EMs) on line 396.

Minor points:

1.      On page 1, lines 8-10, the statement that defines the role of EVs in facilitating intracellular communication among neighboring cells may potentially confuse non-specialist readers, as EV-mediated intercellular communication also involves distant cells.

2.      Abbreviation for TNF-related apoptosis-inducing ligand (TRAIL) appears twice in the article.

3.      On line 242: typo at 'SKB3'.

4.      On lines 283-285: the sentence is confusing.

Minor corrections to grammatical errors are required.

Author Response

Response to Reviewer 2

Major points:

Point 1. On page 6, lines 202-203: Please elaborate on how TRAIL is successfully transfected into the cells. Is TRAIL naturally incorporated into the exosome through simple transfection without any modifications?

Response: Thank you for noting this and providing the feedback for clarification. We have modified the statement in the paragraph to bring this clarity. The mechanism of TRAIL expression by transduction of lentiviral expression vector has now been included in the manuscript with the following statement.

“TRAIL-containing exosomes were developed by transducing TRAIL with k562 K562 leukemic cells with TRAIL lentivirus expression vector [47].

Point 2. On page 7, lines 225-231, these sentences appear rather confusing and require clarification. The abbreviation for interleukin-3 (IL-3Rα) appears to be incorrect.

Response: The sentences in lines 225-231 were re-written and the abbreviation for interleukin-3 receptor (IL-3Rα) was corrected. The modified sentence is as follows.

“Similarly, the N-terminal of LAMP2B fused with interleukin-3 receptor (IL-3Rα) improved exosome targeting efficiency in treating chronic myeloid leukemia (CML) [51]. The IL-3Rα-exosomes, derived from CML cells with highly expressed IL-3Rα, were further loaded with breakpoint cluster region (BCR)-ABL siRNA and imatinib. These exosomes highly accumulated at the tumor site and showed an intense anti-tumor effect with an increased survival rate of xenografted mice.”

Point 3. On lines 243-246: The function of tLyp-1 (tumour homing & penetrating peptide CGNKRTR) and how it enhances exosome delivery into lung cancer should be elaborated.

Response: The function of tLyp-1 in enhancing exosome delivery into lung cancer was elaborated. The modified sentence is as follows;

“These tLyp-1-siRNA exosomes showed enhanced delivery into lung cancer by selectively targeting neuropilin receptors (NRP1 and NRP2) expressed on the tumor tissues”.

Point 4: It would be advantageous to divide Section 3 into smaller subsections since it contains a substantial amount of information presented in a single section.

Response: The authors would like to thank the reviewer for suggesting dividing the section into smaller subsections. We have reorganized the contents of this section within the subsections. The following subsection headlines were introduced to bring clarity and ease for readers.

3.1 EVs Carrying Immune Receptor and Ligand Protein

3.2 EV Signature Protein Fusion

3.3 Genetic Engineering Cancer Cell-derived EVs

Point 5: Is the subsection ‘4.3 Physical Methods’ appropriate to be included in ‘Section 4: Chemical Conjugation Strategies for EV Surface’?

Response: Thank you for noting the error. This section is a separate section. The error in the numbering of the physical methods section was changed to section “5. Physical Methods”.

Point 6. It would be good to create a table summarizing the pre-clinical and clinical trials of the exosome-based therapy being developed as a cancer immunotherapy treatment.

Response: We appreciate this suggestion from the reviewer. We have included a new section and table summarising pre-clinical and clinical studies using exosome-based cancer immunotherapy in Table 2. Since this review article focuses explicitly on surface-engineered extracellular vesicles, we have limited the new section and Table-2 to the studies using surface-modified exosomes.

Point 7. Please define exosome mimetics (EMs) on line 396.

Response: In response to the reviewer’s suggestion, we have defined exosome mimetics at the first instance of discussion in the preceding sentence. The following changes were incorporated into the manuscript;

 “Another study engineered U937 monoblastic cells with anti-PSMA peptide (WQPDTAHHWAT) to generate exosome mimetics (EMs). These EMs presented an anti-PSMA peptide that targets advanced prostate cancer (PC).”

Minor points:

Point 1. On page 1, lines 8-10, the statement that defines the role of EVs in facilitating intracellular communication among neighboring cells may potentially confuse non-specialist readers, as EV-mediated intercellular communication also involves distant cells.

Response: We have re-worded this sentence and also modified the simple summary for non-specialist readers.

The sentence was modified to “Extracellular vesicles facilitate the transportation of biomolecules, such as protein, RNA, and DNA fragments, to communicate with neighboring and distant cells.”

Point 2. Abbreviation for TNF-related apoptosis-inducing ligand (TRAIL) appears twice in the article.

Response: The repeated abbreviation for the TNF-related apoptosis-inducing ligand (TRAIL) was removed.

Point 3. On line 242: typo at ‘SKB3’

Response: This typo is corrected now as SKB3 tumor cells.

Point 4. On lines 283-285: the sentence is confusing. Comments on the Quality of English Language Minor corrections to grammatical errors are required.

Response: Based on the suggestion of the reviewer, the sentence was rephrased as below;

In particular, Ran et al. 2020 fused myostatin propeptide with the second extracellular loop of CD63, which increased exosome serum stability and its delivery efficacy in the MDX mice.

Round 2

Reviewer 2 Report

The authors have adequately responded to most of my concerns, and I recommend this paper for publication in Cancers.

Minor point: I believe the accurate name for the breast cancer cell line is SKBR3, not SKB3.